# Absorption Spectra Description for T-Cell Concentrations Determination and Simultaneous Measurements of Species during Co-Cultures

**DOI:** 10.3390/s22239223

**Published:** 2022-11-27

**Authors:** Bruno Wacogne, Naïs Vaccari, Claudia Koubevi, Marine Belinger-Podevin, Marjorie Robert-Nicoud, Alain Rouleau, Annie Frelet-Barrand

**Affiliations:** 1FEMTO-ST Institute, University of Bourgogne Franche-Comté, CNRS, 15B Avenue Des Montboucons, 25030 Besançon, France; 2INSERM CIC 1431, Besançon University Hospital, 2 Place Saint-Jacques, 25030 Besançon, France; 3Smaltis, Bioinnovation, Rue Charles Bried, 25030 Besançon, France

**Keywords:** T-cell culture, co-culture monitoring, white light spectroscopy, advanced therapy medicinal product

## Abstract

Advanced Therapy Medicinal Products are promising drugs for patients in therapeutic impasses. Their complex fabrication process implies regular quality controls to monitor cell concentration. Among the different methods available, optical techniques offer several advantages. Our study aims to measure cell concentration in real time in a potential closed-loop environment using white light spectroscopy and to test the possibility of simultaneously measuring concentrations of several species. By analyzing the shapes of the absorption spectra, this system allowed the quantification of T-cells with an accuracy of about 3% during 30 h of cultivation monitoring and 26 h of doubling time, coherent with what is expected for normal cell culture. Moreover, our system permitted concentration measurements for two species in reconstructed co-cultures of T-cells and *Candida albicans* yeasts. This method can now be applied to any single or co-culture, it allows real-time monitoring, and can be easily integrated into a closed system.

## 1. Introduction

### 1.1. Context

Advanced Therapy Medicinal Products (ATMPs) are drugs based on genes, tissues, or cells for human use for the treatment of chronic, degenerative, or life-threatening diseases [1]. Genetic modification or tissue engineering give them new physiological, biological characteristics, or reconstruction properties. However, complex and expensive technologies of cell sorting, amplification, genetic transduction, and activation are required to produce these drugs. The whole process takes place in a controlled environment and numerous quality controls are performed throughout the production for up to 10 days. Consequently, the price of these promising therapeutic products restricts the possibility to democratize their use for the greatest number of people. Devices developed during the last few years are not optimal because they do not include/allow online tracking technologies. Only a few parameters such as temperature, pH, or dissolved O_2_ are monitored using sterile probes placed inside the bioreactor. The PAT project (Process Analytical Technology) was born from this observation by the FDA in 2004. This project encourages research and development of new analysis technologies allowing real-time monitoring of all production stages of biopharmaceutical drugs. Concerning ATMPs, the whole production process is quite complex [2], and the above-mentioned quality controls are frequently performed, especially during the expansion phase [3,4]. Multiplying these controls, and therefore samplings, increases the risk of new contaminations.

Therefore, there are two needs: (i) to develop monitoring solutions easily transferable in a closed-loop system for real-time cell concentration measurements without sampling bioreactor content, and (ii) to monitor simultaneously concentrations of several species during their growth and to follow the development of possible contaminations and more generally to monitor co-cultures.

### 1.2. Commercial Availabilities for Cell Counting

Cell concentration has been historically determined by direct measurement of cell number under microscopes through Malassez cells. This most well-known technique is still used with drawbacks of difficulties for visual and manual counting and poor reproducibility due to the relatively small cell volumes sampled and therefore less representative of the culture flask. Alternative and commercial automated methods are now available to facilitate cell counting. Automatic cell counters are commercially available. The LUNA^TM^ system (LOGOS BIOSYSTEMS; [5,6,7]) requires 10 µL of cell suspension and is based on conventional imaging and processing. Other systems developed by IPRASENSE are based on lensless imaging [8] in which cell diffraction figures on a large area are recorded and analyzed to assess cell concentration at a higher accuracy; among them, NORMA uses 10 µL, whereas CYTONOTE is preferred for measurement of adherent cells on larger volumes. In addition, INCUCYTE^®^ (SARTORIUS) used for both adherent and non-adherent cells [9,10] and the HoloMonitor^®^ system (PHI [11]) are in situ microscopy systems based on holographic imaging placed within an incubator. It allows for cell counting directly within different volumes including 96 well plates for high throughputs and/or multiple simultaneous experiments.

Despite their easy use, these commercial systems seem difficult to be integrated into a closed-loop and real-time environment.

### 1.3. Other Biological and Physical Techniques for Cell Qualifications

Other methods can also be used for both cell and subcellular entity qualifications. Some are based on the capture of the biological entity on the biosensor surface by a ligand–analyte reaction such as an enzyme-linked immunosorbent assay (ELISA) [12,13], Surface Plasmon Resonance [14,15], and Quartz Crystal Microbalances [16,17]. However, these methods require a biological interface and a regular regeneration of the surface, which makes transposition to a real-time measurement system difficult. Other methods can be used without a bio-chemical interface. Among them, impedance spectroscopy (or dielectric spectroscopy) has been widely used to study cell culture processes, particularly in the monitoring of mammalian cells [18]. This technique allows cell quantification thanks to their polarization after the application of an alternating electric field and presents several advantages such as in situ analysis of cell culture and rapid measurements. However, this method requires calibration, and the accuracy decreases during the stationary phase of growth [19]. Different spectroscopic methods have been applied for the characterization of mammalian cell culture [20]. Among them, Raman spectroscopy performed either in situ [21] and/or through surface-enhanced Raman scattering [22,23] has already been used for biological purposes [24] during quality controls carried out on cell culture [23] and for pathogen detection [25]. However, the fine and precise data obtained by these techniques may not be required for cell monitoring. Flow cytometry can also be employed for cell counting [26,27] and activation detection [28]. Depending on the optical detection scheme, counting and assessing biological properties for quality control could be performed simultaneously. It is also the case for most of the techniques described above that also allow for simultaneous detection of different species in co-cultivation but could require additional sample treatments such as fluorescence labeling [29].

Indeed, cell counting methods described above all imply considering cells one by one to assess cell concentrations and most require sampling of small volumes poorly representative of what occurs in the bioreactor. Concerning co-culture studies, label-free techniques should be preferred.

### 1.4. Global Methods and Co-Culture Investigations

Measurements without sampling are possible either by derivation or using sterilized optical probes as proposed in [3,4]. Such methods concern the global “light-culture” interaction rather than the behavior of individual particles. Absorption-based methods like turbidimetry or Beer–Lambert law derived techniques have usually been used and preferred for smaller biological entities such as bacteria but can also be applied to mammalian cells [30]. These techniques could also be performed in larger volumes [3,31,32,33,34] to determine cell density and viability [35].

Methods of concentration determination relying on the estimation of only one parameter (Beer–Lambert derived methods or cell counters) cannot be used to simultaneously monitor concentrations of several species. For this, a multi-parameters method should be employed. The detection of several species has already been reported but remains a challenge. Non-optical methods such as quartz crystal sensors [36] or electrochemistry [37] have recently been used either for bacteria detection in complex fluids or multiple bacteria detection. White light spectroscopy and light scattering analysis have already been used for the detection of bacteria in co-culture [38,39]. We previously determined B-cell concentration using white light spectroscopy and its use to detect contaminations [3,4]. The use of fiber optic Fourier Transform Infra-Red spectroscopy has also been reported [40]. Only a few papers mention both cell monitoring and contaminant detection; in particular, advanced signal processing applied to Raman spectroscopy has been proposed [41]. Together with normal operation condition monitoring, the authors demonstrated the detection of growth perturbations 5 h after the discontinuation of cell feeding and detected the effects of contamination with their monitoring algorithm. However, the nature of the contamination and the time required to detect it were not specified.

### 1.5. Current Needs and Proposed Method

To summarize, there are needs for an online and sampling-free cell concentration monitoring device, and for methods allowing simultaneous concentration measurements for several species. Because they are contactless, optical techniques are good candidates to meet these needs. Since each biological species exhibits its absorption spectrum, optical spectroscopy should allow for discriminating spectral signatures of species during co-culture.

In this paper, white light spectroscopy was used to measure T-cell concentrations from the shape of the absorption spectra of different dilutions. Indeed, measuring concentrations from the spectral value at only one wavelength (Beer–Lambert derived methods) we proposed [30] cannot be used to measure concentrations of several species simultaneously. Indeed, the shape of the whole absorption spectrum of a mixture is a combination of the shapes of each individual species. Therefore, mathematical treatment of the mixture spectrum allows the measurement of concentrations of individual species simultaneously. The paper is structured as follows: Section 2 presents the materials and methods used in this study. Numerical and experimental results concerning the spectral shape analysis and the possibility to extend this method in the case of two simultaneous concentration measurements are the subject of Section 3. Results will then be discussed (Section 4), and conclusions presented (Section 5).

## 2. Materials and Methods

### 2.1. CEM Preparation

CEM cells (ATCC^®^ CRL-2265TM) are T lymphoblasts that were supplied by the French Blood Agency (EFS Etablissement Français du Sang). They were grown in RPMI-1640 medium (P04-16515, PAN-Biotech^®^, Germany) supplemented with 25 mM HEPES (P05-01500, PAN Biotech^®^, Germany), 10% heat-inactivated FBS (10270 -106, Fischer Scientific^®^, France), and 1% penicillin (10 kU/mL^−1^)/streptomycin (10 mg.mL^−1^) (FG101-01, TransGen Biotech^®^, China). The cells were maintained at 37 °C in a humidified atmosphere containing 5% CO_2_.

Different concentrations were prepared by diluting cuvettes in RPMI medium to obtain concentrations between 10^5^ and 10^6^ cells.mL^−1^. To generate a robust spectroscopy model, a large number of different spectra is required (i.e., a large number of associated concentrations). Since each spectroscopy measurement required 2.5 mL of cell solution, 8 weeks of cell culture were necessary. Each week, diluted cuvettes of 8 different concentrations distributed between 10^5^ to 10^6^ cells.mL^−1^ were prepared resulting in 80 experimental data. Three cell counts (using the LUNA-II Automated Cell Counter, Logos Biosystems^®^, South Korea, supplier France, with trypan blue V/V, 15,250,061 Fisher Scientific^®^, France, with 10 µL of cell suspension) and one spectral measurement were performed with each cuvette for mathematical modeling purposes.

### 2.2. Cultivation of CEM Cells over 30 Hours

Three days post passage CEM cells were centrifugated at 500× *g* for 10 min at room temperature. The pellet was resuspended at a concentration of 5 × 10^5^ cells.mL^−1^. They were maintained at 37 °C for 30 h. Spectral measurements were performed every hour for the first 4 h, every 30 min from T = 4 to T = 11 h, and every 2 h between T = 21 and T = 30 h.

### 2.3. Concentration Ranges for Optical Absorption Modeling of Candida Albicans

The *Candida albicans* yeast strains (ATCC10231) were grown on SAB plates (PO5001A, OXOID, France) prior to liquid culture in SAB (TV5054E, Oxoid, France) aerobically at 22.5 °C at 200 rpm for 2 days. They were recovered by two centrifugations at 10,000× *g*, 10 min, 20 °C, and resuspended in PBS 1x pH7.4 (Sigma, USA). The optical density of the re-suspension was measured in a spectrophotometer at 600 nm (BIOWAVE DNA, BIOCHROM, United Kingdom). Afterwards, different yeast concentrations from 0.5 × 10^6^ to 4 × 10^6^ cells.mL^−1^ were prepared for experiments and analysis.

### 2.4. Spectroscopic Absorption Measurements

Spectral absorption measurements of CEM suspensions were performed using the experimental setup shown in Figure 1 (adapted from [30]). The spectroscopy measuring system consists of a light source (AvaLight-DH-S-BAL, Avantes^®^, the Netherlands, supplier France) connected by optical fibers (Thorlabs, USA, supplier France, M25L01) to a cuvette holder (Avantes, the Netherlands, supplier France, CUV-UV/VIS). The white light source was switched on about 30 min before measurements to allow temperature and spectral characteristics’ stabilization. After propagation through the cuvette, the light was transmitted to the spectrophotometer (Ocean Optics, USA, supplier France, USB 4000 UV-VIS-ES) for spectra acquisition. Before each measurement, a reference spectrum was acquired using a cuvette containing RPMI medium. Suspensions were homogenized by several gentle inversions before each spectroscopy measurement. Spectra were recorded in transmission, in the wavelength range between 177 nm and 892 nm with a step of 0.22 nm using the OceanView (Ocean Insight, USA, supplier France,) software.

### 2.5. Spectral Data Processing

The 80 CEM spectral data were recorded into a text file and then transposed to Excel. The data obtained in transmission were converted into absorption percentages, and all calculations were performed using Matlab^TM^ R2020b software, USA, supplier France. Only wavelengths between 330 nm and 860 nm were considered to remove measurements with high background noise. Artifacts due to energetic emission peaks of the deuterium lamp were numerically removed. Regularly, absorption spectra of neutral densities (THORLABS, USA, supplier France, NE05B and NE10B) were recorded and compared to the supplier’s data to ensure correct absorption spectra measurements. A home-developed spectra quality estimator (unpublished data) was used to remove badly shaped spectra. A total of 75 CEM spectra were used in this paper. The same protocol was used for *Candida albicans* (hereafter CA) spectra. Seven CA spectra were used in this study for demonstration purposes. In this study, the absorption spectra $Abs_{species}\left(\lambda ,C\right)$ are defined as:(1)$$Abs_{species}\left(\lambda ,C\right)=100\left(\frac{1}{T_{species}\left(\lambda ,C\right)}\right)$$

Here, $T_{species}\left(\lambda ,C\right)$ is the transmittance of the corresponding species.

CEM and CA co-cultures were not performed in this study. The proof of concept of simultaneous co-cultured species concentrations’ measurements uses artificially computed spectra from experimental spectra of CEM and CA. They were computed based on the additivity law of absorbances (or Optical densities OD). From this law, and combining definitions of absorbance and transmittance, it can be shown that the absorption spectrum of a mixture of ‘n’ different species is given by Equation (2):(2)$$Abs_{Mix}\left(\lambda ,C_{1}\ldots C_{n}\right)=100\left\{1-\prod_{i=1}^{n}\left(1-\frac{Abs_{i}\left(\lambda ,C_{i}\right)}{100}\right)\right\}$$

Equation (2) was used to calculate the spectra of CEM and CA mixtures.

## 3. Results

First, the method to mathematically describe CEM cell and *Candida albicans* (hereafter CA) absorption spectra was established. Then, CEM concentration spectral monitoring during a 30-h cultivation experiment using shapes of the absorption spectra was performed. In the end, the shape of the absorption spectra of CEM/CA co-culture was reconstructed, and concentrations of each species were calculated using the function describing the global absorption spectra of mixtures.

### 3.1. Modeling CEM Absorption Spectra

Absorption spectra of dilution ranges of CEM cells prepared as explained in Section 2.1 were measured using the experimental setup described in Section 2.4, resulting in Figure 2.

In the Beer–Lambert derived model [30], the value of the maxima of each spectrum was used to compute the corresponding concentration, but this model cannot be used to simultaneously compute concentrations of species within a mixture. Here, the information contained in the whole absorption spectrum was exploited to establish a model describing the evolution of CEM absorption spectra with the concentration. It was based on the observation that each spectrum from Figure 2 could efficiently be fitted using two Gaussian functions with a fitting R^2^ of the order of 0.98 (data not shown). The absorption spectrum can then be written as follows:(3)$$Abs_{CEM}\left(\lambda ,C\right)=\sum_{i=1}^{2}a_{i}\left(C\right).exp\left\{-{\left(\frac{\lambda -b_{i}\left(C\right)}{c_{i}\left(C\right)}\right)}^{2}\right\}$$

Here, $a_{i}\left(C\right)$, $b_{i}\left(C\right),$ and $c_{i}\left(C\right)$ are the amplitude, the position, and the width of Gaussian ‘i’.

Note that the amplitudes, the positions, and the widths of the Gaussian functions depended on the concentration C. They were called sub-functions of the Gaussians. The six sub-functions needed to be mathematically determined. The next step was to determine what equations described them using fitting iterations explained below.

#### 3.1.1. Iterated Fittings Approximation

The first fitting step consisted of directly fitting spectra with two Gaussians letting the quantities $a_{i}\left(C\right)$, $b_{i}\left(C\right)$, and $c_{i}\left(C\right)$ free (Figure 3a). The values of the Gaussian coefficients were plotted as a function of the concentration. Figure 3a suggested that the center of Gaussian 1 could be considered constant and equal to 496.9 nm.

A second fitting iteration considering $b_{1}=496.9$ was then conducted (Figure 3b) where $a_{2}$ can be considered constant and equal to 23%. Iterations were repeated until no sub-function could be considered constant anymore (Figure 3e).

At this stage, the CEM spectral shape was written as follows, with one constant Gaussian function and a variable Gaussian function in which amplitude and width depended on CEM concentration:(4)$$Abs_{CEM}\left(\lambda ,C\right)=a_{1}\left(C\right).exp\left\{-{\left(\frac{\lambda -b_{1}}{c_{1}\left(C\right)}\right)}^{2}\right\}+a_{2}.exp\left\{-{\left(\frac{\lambda -b_{2}}{c_{2}}\right)}^{2}\right\}$$

Note that, at the iteration #5 stage, data representing $a_{1}\left(C\right)$ and $c_{1}\left(C\right)$ were much less dispersed than they were at iteration #1. The next step was to mathematically describe sub-functions $a_{1}\left(C\right)$ and $c_{1}\left(C\right)$. This was done by fitting $a_{1}\left(C\right)$ with an exponential function and $c_{1}\left(C\right)$ with a power function (Figure 4).

A first approximated CEM spectra shape function could then be written as follows with numerical approximated parameters given in Table 1:(5)$$Abs_{CEM}\left(\lambda ,C\right)=100.\left(1-10^{-p1a1.C}\right).exp\left\{-{\left(\frac{\lambda -b_{1}}{p1c1.C^{p2c2}}\right)}^{2}\right\}+a_{2}.exp\left\{-{\left(\frac{\lambda -b_{2}}{c_{2}}\right)}^{2}\right\}$$

These parameters were approximated since they were obtained sequentially using iterated fittings. Successive fittings allowed for identifying constant sub-functions and making explicit the concentration dependency of non-constant sub-functions.

#### 3.1.2. Parameters Calculation Using a Minimization Method

CEM spectra evolution with concentration formed a surface that could then be directly adjusted with Equation (5) by fitting parameters simultaneously. The “fminsearch” function (Matlab^TM^, documentation available at: https://fr.mathworks.com/help/matlab/ref/fminsearch.html, last access on 28 October 2022) was used to determine the set of parameters that minimized the following error function:(6)$$error=\sum_{\lambda }\sum_{C}{\left(Abs_{CEM}\left(\lambda ,C\right)-ExpSpectra\right)}^{2}$$

Here, ExpSpectra represented the 75 absorption spectra shown in Figure 2.

The “fminserach” function required a set of starting points. Approximated parameters given in Table 1 were used as starting points to minimize Equation (6) (Figure 5 and final parameters were calculated with the minimizer algorithm in Table 2).

Examples of fittings for two experimental spectra with different concentrations were performed (Figure 6) using Equation (5) and parameters from Table 2. The fixed Gaussian could then be considered as a sort of baseline appearing for large wavelengths.

Parameters displayed in Table 2 differed from those given in Table 1. Then, the comparison of R^2^ values from fitting (Table 1) and minimization (Table 2) were compared (Figure 7).

Both sets could describe the experimental CEM spectra efficiently (Figure 7) since the R^2^ values obtained when fitting experimental spectra with Equation (5) using either parameters from fitting or from minimization were quite similar, with slightly higher R^2^ values with the minimization algorithm. Moreover, it seemed that the shape fitting was inefficient for extreme concentration values. The CEM spectral model given by Equation (5) was used to compute concentrations and to compare results to measurements performed with the automatic cell counter.

#### 3.1.3. Measuring CEM Concentrations Using the Shape of the Absorption Spectra

Then, the accuracy with which the model calculated CEM concentrations was studied. First, a rapid estimation of the model accuracy was established, and descriptors of the accuracy were defined. Second, cross-validation of the model was performed to estimate the accuracy more realistically.

Global evaluation of measurement performances.

Equation (5) was used to fit spectra from Figure 2 to calculate CEM concentrations and resulted in Figure 8a. It could be observed that, despite a low R^2^ value, the model could accurately compute CEM concentrations at extreme concentration values.

Ideally, calculated concentrations should be situated on the *Y* = *X* line (dashed black in the Figure). Figure 8b was used to define descriptors of the accuracy of the spectral concentration measurements. Fitting the experimental data with the function: Y=X+Bias is shown as a magenta dashed line in Figure 8b.

The “Bias” was an estimation of how different calculated concentrations were from the counter values. A positive “Bias” means that the model overestimated the concentration while a negative one shows that the model underestimated the concentration.

The dispersion of the fitting results around the Y=X+Bias line was represented by the large green area (Figure 8b). The width of this green line was called “Disp.” (dispersion). The larger the “Disp.” could be, the less accurate the measurements would be. The dispersion could be calculated using a modified form of the Standard Deviation definition as shown in Equation (7):(7)$$disp=\sqrt{\frac{1}{n}\sum_{i=1}^{n}{\left(C_{i}^{calc}-\left(C_{i}+Bias\right)\right)}^{2}}$$
where $C_{i}$ was the counter value of the sample number ‘*i*’ and $C_{i}^{calc}$ the corresponding calculated concentration.

The dispersion and bias values of the experimental results (Figure 8a) were given by:

Disp. = 5.6 × 10^4^ cells (about 9% at center concentration range).

Bias = 883 cells (virtually zero).

The virtually zero bias indicated that the model did not over/under-estimate the counter values. The accuracy was here less than 10%, whereas an accuracy of about 20% is still acceptable (personal communication with the French Blood Agency). However, this accuracy was relatively good because the model was tested on data used to establish it. The next step was to evaluate the model by conducting cross-validation.

Cross-validation evaluation.

A first model was established with five experimental sets called “model data”. This model consisted of Equation (5) with parameters calculated with the five chosen model sets. This model was then applied to the three remaining data sets called “test data”. The bias and dispersion values were calculated for this first combination. This process was iterated for all possible combinations. Figure 9 shows the dispersion and bias values for all possible “model” and “test” set combinations.

The average dispersion was equal to 5.2 × 10^4^ cells (dashed black line in Figure 9). It was slightly higher than what was calculated globally and represented about 8.7% at the center concentration range (still acceptable). The bias results were more surprising since the bias value decreased with the combination number as detailed in Figure 10.

Positive bias values corresponded to “model data” mostly recorded during the 4 first weeks (W11 to W17) and “data sets” mostly recorded during the last 4 weeks (W18 to W23), and vice versa for negative bias values. The average bias over the period was 622 cells, the same virtually zero value as the one calculated globally.

The accuracy of the CEM spectral model was about 13%. This value will be discussed later regarding the dispersion due to plastic spectroscopy cuvettes. The model was applied to monitor the evolution of the concentration during a 30-h CEM cultivation experiment.

#### 3.1.4. Measuring CEM Concentration over 30 Hours

The evolution of the concentration of CEM cells cultivated in a single spectroscopy cuvette was measured using the CEM spectra shape model (Figure 11).

Since the experimental set-up was not yet automated, no data were recorded during the night. The spectra shape model calculated an initial CEM concentration of 4.6 × 10^5^ cells.mL^−1^ while the cuvette was initially filled with a concentration of 5 × 10^5^ cells.mL^−1^. This value was correct, especially considering that the cell counter measurements were not fully representative of the real concentration within the culture flask (see Discussion and [30]).

Black circles (Figure 11b) showed concentrations calculated using the Beer–Lambert derived model [30]. Results were similar to calculations obtained with the present shape model, but a slight underestimation was observed with the Beer–Lambert derived model after 20 h, which seemed to increase with time.

While fitting the evolution of CEM concentration with an exponential function, a doubling time was determined of about 24 h 35 considering data over the 30-h experiment. This corresponds to what was expected. A doubling time of 20 h was measured during the first 11 h of the experiment while a doubling time of 27 h 42 was measured during the last 10 h. This will be discussed later.

Results obtained during this 30-h cultivation experiment showed an evolution of the CEM population in accordance with what was expected. Data seemed to be less dispersed than what could have been expected from Section 3.1.3. Indeed, because the experiment was conducted in a single spectroscopy cuvette, the dispersion due to the cuvette did not exist here. The dispersion was measured with respect to the exponential fitting as follows:(8)$$disp_{\Delta t}=\sqrt{\frac{1}{n}\sum_{i=1}^{n}{\left(C_{calc}\left(t_{i}\right)-ExpFit_{\Delta t}\left(t_{i}\right)\right)}^{2}}$$

Here, Δt referred to the period during which the dispersion was measured (i.e.,the first 11 h, the last 10 h, and the entire 30-h experiment period), $C_{calc}\left(t_{i}\right)$ the concentration calculated at time $t_{i}$, and $ExpFit_{\Delta t}\left(t_{i}\right)$ the value of the exponential fitting during the Δt period at time $t_{i}$. The following dispersions were obtained:Disp(0–11)h = 1.9 × 10^4^ cells;Disp(21–30)h = 2.2 × 10^4^ cells;Disp(0–30)h = 1.8 × 10^4^ cells.

Whatever the period, the dispersion was around 2 × 10^4^ cells (i.e., 3.3% at the center range), much lower than the values obtained using multiple cuvettes for model establishment.

To summarize, the spectral model allowed for accurate calculation of CEM concentration. The next section presents illustrations of simultaneous concentration measurements in co-cultures.

### 3.2. Measuring Concentrations with Reconstructed Co-Cultures

The global spectral shape function described in Equation (2) could also be used to analyze reconstructed spectra of a mixture of CEM cells and *Candida albicans* yeasts without co-culture experiments. It was first necessary to determine the spectral shape function of *Candida albicans.*

#### 3.2.1. Establishing the Candida Albicans Spectra Shape Equation

CA spectra shape equation was established using the same method used for CEM cells (Section 3.1.1 and Section 3.1.2) generating Figure 12.

The spectra shapes of CA yeasts (Figure 12) were similar to the spectra of CEM cells (Figure 2). The main stages of the spectra shape function construction are summarized below:
CA spectra could efficiently be fitted with two Gaussian functions;Only 4 iterated fittings were required resulting in the existence of three sub-functions describing the evolutions of $a_{1}\left(C\right)$, $b_{1}\left(C\right),$ and $a_{2}\left(C\right)$. Each of them could be fitted with a logarithm function;The final parameters obtained with the minimization algorithm are given in Table 3.

Equation (9) represented the CA spectrum shape function:(9)$$\begin{array}{l} Abs_{CA}\left(\lambda ,C\right)=\left(p1a1+p2a1.log_{10}\left(C\right)\right).exp\left\{-{\left(\frac{\lambda -\left(p1b1+p2b1.log_{10}\left(C\right)\right)}{c_{1}}\right)}^{2}\right\}\\ +\left(p1a2+p2a2.log_{10}\left(C\right)\right).exp\left\{-{\left(\frac{\lambda -b_{2}}{c_{2}}\right)}^{2}\right\} \end{array}$$

The CA model allowed for the determination of dispersion and bias values:Disp. = 2 × 10^5^ cells (about 10% at the center concentration range);Bias = 7804 cells.

#### 3.2.2. Examples of Double Concentration Measurements

Examples of reconstructed CEM/CA co-culture were then calculated using Equation (2), and both concentrations were fitted using Equation (2) written as follows:(10)$$Abs_{Mix}\left(\lambda ,C_{CEM},C_{CA}\right)=100\left\{1-\left(1-\frac{Abs_{CEM}\left(C_{CEM}\right)}{100}\right).\left(1-\frac{Abs_{CA}\left(C_{CA}\right)}{100}\right)\right\}$$

Examples of reconstructed spectra and simultaneous concentration measurements are shown in Figure 13.

CEM and CA concentrations were calculated considering all possible combinations of species concentrations (i.e., 450 combinations). Bias and dispersions related to each combination were computed:CEM: disp. = 1.28 × 10^5^ cells (21% at the center range);CEM: bias = 3.2 × 10^4^ cells;CA: disp. = 5.8 × 10^5^ cells (29% at the center range);CA: bias = −1.6 × 10^5^ cells.

Dispersions and bias were quite large because of two factors. First, cuvettes and counter dispersions were added when reconstructing CEM/CA mixtures. This would not be the case for real co-cultures in a single cuvette as mentioned in Section 3.1.4. Second, a crosstalk (influence of CA presence on CEM measurements and vice versa) existed between CEM and CA measurements as shown in Figure 14 where CEM and CA calculated concentrations were plotted as a function of species concentrations in the reconstructed spectra.

Colored planes in the figures represented the mean evolution of the species concentrations calculated in the 3D representations. Equations of the mean planes were written as follows:(11)$$\tilde{P_{CEM}}\left(C_{CEM},C_{CA}\right)=p0_{CEM}+p1_{CEM}.C_{CEM}+p2_{CEM}.C_{CA}$$
(12)$$\tilde{P_{CA}}\left(C_{CEM},C_{CA}\right)=p0_{CA}+p1_{CA}.C_{CEM}+p2_{CA}.C_{CA}$$

Parameters are given in Table 4.

Ideally, $p0_{CEM}$, $p2_{CEM}$, $p0_{CA}$, and $p1_{CA}$ should be equal to zero, but, here, the crosstalk was revealed by the non-zero values of $p2_{CEM}$ and $p1_{CA}$. This aspect is discussed below.

## 4. Discussion

### 4.1. Format of the Spectroscopy Data

In our study, optical spectra were expressed in absorption measured in percentage, but investigations could have been conducted in any other equivalent formats since absorbance (or OD), transmission, or transmittance spectra all strictly contain the same information. Our goal was to provide a method to measure concentrations in-line, without sampling, close to or inside a bioreactor. To this end, compact and low-cost components must be chosen for this purpose. We did not consider methods based on the use of ultra-sensitive detectors such as Photomultiplier Tubes usually used in plate readers. Therefore, data corresponding to low transmission (i.e., high OD) were not fully reliable in our case. Therefore, we decided not to consider absorbance measurements. The choice between transmission and absorption was made considering that absorption spectra were easier to mathematically describe than transmission spectra.

### 4.2. Ways to Optically Measure Concentrations

With the present study and our previous one, three optical methods are now available to measure cell concentrations: considering the value of the absorption maxima (Beer–Lambert derived model [30]), considering the shape of the absorption spectra using iterated fittings, or a minimization algorithm. All three are roughly equivalent. However, although equivalent (Figure 11b), the Beer–Lambert derived method cannot be employed to measure concentrations of several species simultaneously. Successive fittings or minimization algorithm were also equivalent (Figure 7) with slightly higher R^2^ when spectra were fitted with the minimization algorithm. In any case, both must be considered because the successive fittings produced starting values for the minimization algorithm.

### 4.3. Successive Fittings

Iterated fittings showed that it can be approximated that only the amplitude $a_{1}$ and the width $c_{1}$ of the first Gaussian evolved with the concentration. As already mentioned, the second Gaussian should be considered as a baseline. $c_{1}\left(C\right)$ was described with a power function. This function cannot be related to any light–matter interaction process and another function could have been used instead. $a_{1}\left(C\right)$ was fitted with an exponential function. This function led to the best adjustment efficiency with R^2^ = 0.96. This exponential function was chosen because the evolution of $a_{1}\left(C\right)$ with the concentration was very similar to the evolution of the maximum value [30]. This exponential equation was directly derived from the Beer–Lambert law. Both ‘a’ parameters were very close: 7.5 × 10^−7^ [30] and p1a1 parameter equal to 7.45 × 10^−7^ in the present study. This showed that the concentration related principally to the absorption amplitude. This was also observed by the fact that the aspect of $a_{1}\left(C\right)$ was conserved throughout the successive fittings (to be compared to the evolution of $c_{1}\left(C\right)$ for example). Previously, the position of spectra maxima was shown to evolve with concentration [30]. In the case of Figure 3e only, this should not happen because the centers of both Gaussians were fixed. However, looking at successive iterations (Figure 3), it was clearly seen that $b_{2}$ slightly evolved with the concentration, which was in accordance with what was observed before. This evolution was lost when considering $b_{2}$ as a constant. However, this was compensated by the fact that the dispersion of $c_{1}\left(C\right)$ strongly decreased with successive fittings. Indeed, considering $b_{2}$ not constant would have kept $c_{1}\left(C\right)$ more dispersed, and the overall fitting R^2^ would not have been improved. We decided to simplify the mathematical expression of $Abs_{CEM}\left(\lambda ,C\right)$ by considering $b_{2}$ constant.

The second Gaussian is considered as a baseline with absorption occurring mainly in the near infrared region. CEM concentration could have been measured with only one Gaussian function, but the R^2^ of the fitting would have been lower, possibly leading to a reduced concentration measurement accuracy. Trying to keep R^2^ as high as possible is crucial when considering co-cultures where one of the species exhibits a high absorption around the 800 nm wavelength.

### 4.4. Considerations about Data Dispersion and Bias

Basic plastic spectroscopy cuvettes were used in these experiments, and blues lines (Figure 8a) were related to their dispersion. An ancillary study (results available on demand) was conducted to estimate the effect of cuvette variability. The normal distribution of the absorptions measured at a 600 nm wavelength on 100 cuvettes showed a full width at a half-maximum of ±2.5% corresponding to ±3.8 × 10^4^ cells at center range. Blue lines in Figure 8a represented this cuvette variability. It should be compared to the calculated dispersion observed in Figure 9a related to the cross-validation calculations. Aspects concerning CEM cell concentration measurements using an automatic cell counter should also be considered. This device uses only 10 µL of the suspension to calculate cell concentration by image processing. During monitoring experiments [30], a dispersion of about 10% of the automatic counter result was measured not due to a dysfunction of the cell counter but linked to the small volume hardly representative of the actual cell concentrations within the culture flask or in the spectroscopy cuvette.

Dispersion and bias (Figure 8b) were calculated in a “vertical” way between the black dashed line and the experimental data. The algebraic distance between the black and magenta dashed lines (distance orthogonal to these lines) was not considered since the definition of the dispersion and bias was related to the discrepancy between the concentration optically measured and the concentration measured with the automatic counter. A “horizontal” dispersion also existed due to the above-mentioned remark concerning the automatic cell counter. The dispersion was only 3.3% at the center range while monitoring cell growth within the same cuvette (Figure 11), and this value corresponds to the real method accuracy. This value is sufficient for real applications. For example, counting cells with a Malassez cell leads to accuracies between 10–20% and is human dependent. Commercial cell counters are very accurate, but they require very small volumes (10–20 µL) that are hardly representative of the content of the suspension under investigation. We measured dispersion of about 10% with an automated cell counter (reference [30] of the paper). As already mentioned, “an accuracy of about 20% is still acceptable” (personal communication with the French Blood Agency).

To summarize, dispersion values mentioned in Section 3.1.3 and Section 3.2.1 are mostly linked to cuvette optical properties variability and cell counter representativeness and not to an intrinsic inaccuracy of the spectral measurements. Variability of the cuvette properties is no longer a problem when monitoring cell culture in a single cuvette (3% reported in Section 3.1.4). In addition, because spectral measurements are performed with large volumes (about 70 µL), they are more representative of the suspension content.

One of the most surprising observations was that the bias decreased with the combination number and seemed related to the weeks during which “model sets” and “test sets” were selected along the cross-validation estimation. No clear explanation for this phenomenon has yet been found since this bias decrease seemed not to be correlated to the number of passages that cells have undergone; environmental factors may be responsible (the first experiments were performed in March and the last experiments in June).

### 4.5. Remarks about the Cell Multiplication Times

Concentrations measured from the shape of the absorption spectra were compared to those measured from the Beer–Lambert derived model [30] (Figure 11b). Normally the mammalian cell population doubles every day. A common doubling time value of 26 h is traditionally considered. However, doubling times strongly depend on cultivation protocols [42]. Results shown in Figure 11b suggested that doubling time evolved with time. Apparently, cells divided faster during the first 11 h (doubling time 20 h 00 min) than during the last 10 h (doubling time 27 h 42 min). Overall, the doubling time estimated during 30 h of the experiment was 24 h35, closer to what is usually acknowledged. Two explanations can be proposed: (i) This variation in the doubling time was not yet observed because cell concentration controls were normally not performed every 30 min as it was in the present study. (ii) Cultivation took place in a spectroscopy cuvette with a reduced volume of RPMI medium (2.5 mL), which could have reduced the multiplication rate when the cell concentration became larger. Indeed, available nutrients decreased more rapidly in such small volumes while the CO_2_ level increased more rapidly. Note that measurements were made over 30 h of cultivation, far from the stationary phase, which occurs after several days in normal culture conditions.

### 4.6. Concerning the Reconstructed Co-Cultures CEM/Candida Albicans

Seventy-five CEM spectra were used to construct the shape model, which was therefore quite representative. Conversely, only six dilution ranges of *Candida albicans* were used explaining that the shape model could be less accurate. Moreover, the absorption spectra of yeast and mammalian cells were quite similar. This made the fitting of both species quite difficult and reduced the accuracy of the result because of the crosstalk between each species’ measurements. The parameters given in Table 4 allow an understanding of the effect of shape similarity on the results. Concerning the CEM mean plane, $p2_{CEM}$ was related to the contribution of CA. It should be zero while $p1_{CEM}$ should be equal to 1. This was not exactly the case because $p2_{CEM}$ = 0.04 and $p1_{CEM}$ = 1.029. This means that the CEM calculation was slightly influenced by CA.

We recall that this illustration of co-culture monitoring was performed using reconstructed mixture spectra. The goal was to demonstrate simultaneous concentration calculations from the shape of the absorption spectra. Simultaneous monitoring of CEM cells and *Escherichia coli* bacteria co-culture is currently being investigated, and accuracies of about 3% for both species’ concentration measurements are observed (data not shown).

### 4.7. Position of Our Studies and Model to Others

Initially, our work aimed at the monitoring of cell cultivation and the detection of bacterial contaminations in cell cultures in real time, without sampling and labeling. Indeed, spectroscopy methods offer the possibility to simultaneously measure single and multiple species concentrations in (co)-cultures. Our model was established on spectra shapes and allowed monitoring of single and co-cultures of biological elements and co-determination of concentrations of two different species.

Other studies have been performed using either label-free non-optical techniques or spectroscopies as detailed thereafter and reviewed in [20]. They present different advantages and drawbacks, in terms of high-throughput and online analysis. Using other parameters, they succeeded in the detection of only one element and did not analyze absorption spectra.

Label-free non-optical techniques can be used to study contaminations but most of the time aimed at the study of the presence of one element with another one. Among them, as an example, QCM was used to detect *E. coli* in raw milk [36] with a response time of the sensor of about 4 h, a duration acceptable for real-time detection. However, this technique requires strictly controlling sample temperature and milk was different from the cell culture where cell concentration evolves with time. Electrochemistry was performed with pure bacteria suspensions and enabled a detection limit down to 30 CFU.mL^−1^ [37].

Other techniques based on the application of Beer–Lambert law were reported for co-detection. Some authors tried to detect *Plasmodium berghei* and *Trypanosoma evansi* in mice whole blood by measuring the absorption at a single 650 nm wavelength [38]. They could detect only *Plasmodium* with a high response, already a challenge in whole blood, and reported difficulties in getting signatures below 600 nm because of a too low Signal- to-Noise ratio. Their results could be linked to the difficulty to detect multiple species at only one wavelength and/or to the difference in size and/or composition of both species.

UV/Vis spectroscopy has been described as well-suited for the determination of cell density [20]. Studies within the UV region and more particularly in a reduced wavelength window around 290 nm allowed for performing only cell counting while measuring absorption in 96-well plates, but no multiple detection was achieved by this method, which is halfway between conventional Beer–Lambert law and spectroscopy over a wide wavelength range [31]. Other studies were performed on the full UV-visible range and measured the absorption of Chinese Hamster Ovary (CHO) cells [34,35]. Cell concentrations and viabilities were estimated on the base of Partial Least Squares (PLS) models. These studies require either an optical commercially available probe enabling potential transposition of the technique to an in-line system like ours [35] or were performed in very small volumes (2 µL), which are less representative. In contrast, our model using a wide wavelength range is able to detect a single cell and determine its concentration even in the presence of other elements in large volumes, which are more representative of the cultivation flask.

Only one method developed recently and based on light scattering analysis allowed for simultaneous measurements of two species. The authors succeeded to determine the biomass of two bacteria *Lactococcus lactis* and *Kluyveromyces marxianus* different in size (0.5–1.5 µm for LL and 4–8 µm for KM) in co-culture in 48-well plates [39]. The experimental setup that they proposed, slightly more complex than the one presented here, could also be efficiently transposed in an online and sample-free device. This technique is also influenced by morphological changes and could be as accurate as our model. Since our analysis enabled analysis of co-cultivation of species that exhibit similar sizes (about 10 µm) generating similarity of measured absorption spectra (Figure 2 and Figure 12), i.e., determination of concentrations in extreme cases, it could easily be even more accurate with species with different absorption spectra such as bacteria [3,4].

## 5. Conclusions

This paper highlighted a mathematical description of the shape of the absorption spectra of CEM cells. This description was used to measure CEM cell concentrations as it could have been done using Beer–Lambert-derived methods. Our model allowed for monitoring of a CEM cultivation for over 30 h in a single spectroscopy cuvette with an accuracy of 3.3% and a determination of the cell population doubling rate, even evolving a bit to 24 h 35 min in accordance with what should be expected. In addition, using the shape of the absorption spectra allowed for measuring simultaneously individual species concentrations in the case of co-culture. This accuracy could be increased while analyzing species with more different spectra shapes.

In addition to a high accuracy, the use of white light spectroscopic method presents a big advantage of being easily integrated within a device without sampling, in a closed environment and enables real-time measurements, useful for quality control during ATMP and cell production.

## Figures and Tables

**Figure 1 sensors-22-09223-f001:**
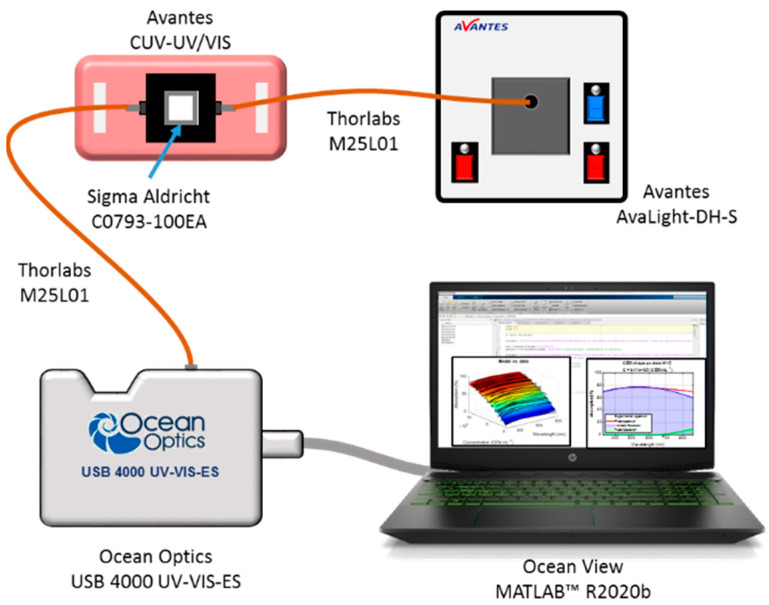
Experimental setup used for measuring absorption spectra.

**Figure 2 sensors-22-09223-f002:**
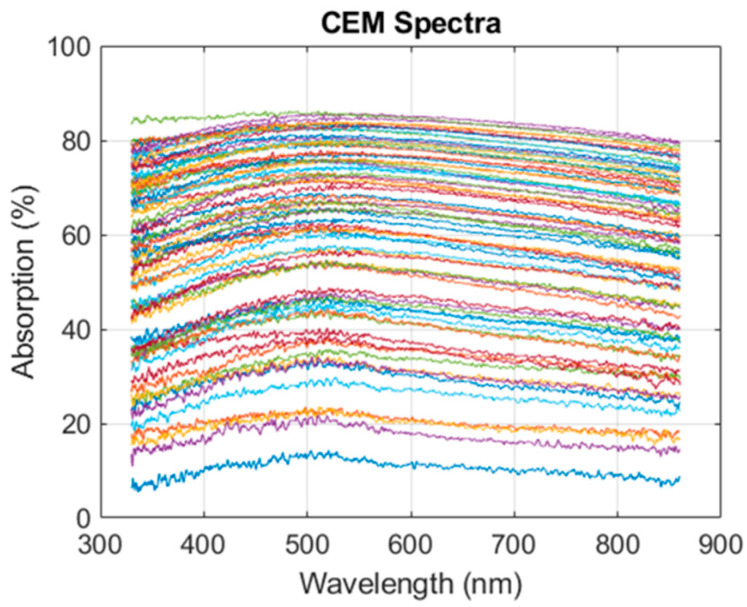
CEM absorption spectra. CEM concentrations range from 7 × 10^4^ to 1.15 × 10^6^ CEM.mL^−1^; *n* = 75 spectra measured over 8 weeks.

**Figure 3 sensors-22-09223-f003:**
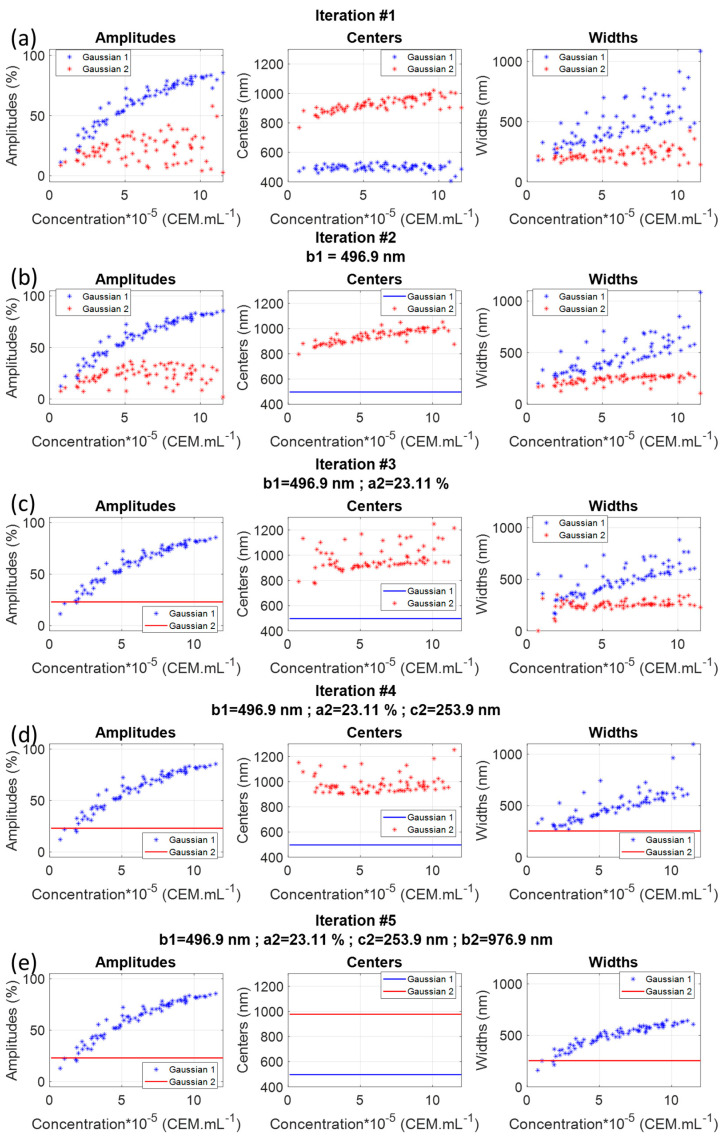
Iterative fitting of experimental CEM spectra with two Gaussian functions.

**Figure 4 sensors-22-09223-f004:**
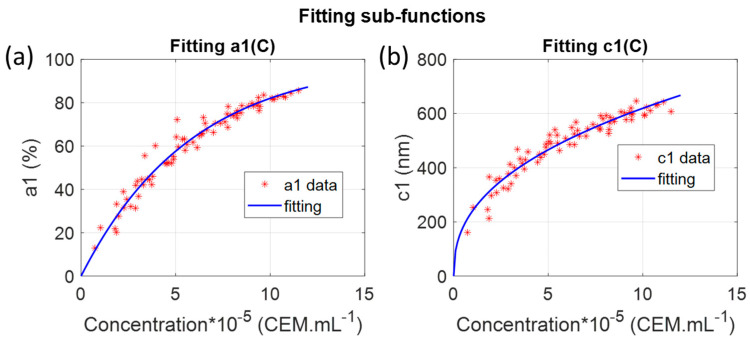
Fitting sub-functions. (**a**) fitting
$a1\left(C\right)$; (**b**) fitting
$c1\left(C\right)$.

**Figure 5 sensors-22-09223-f005:**
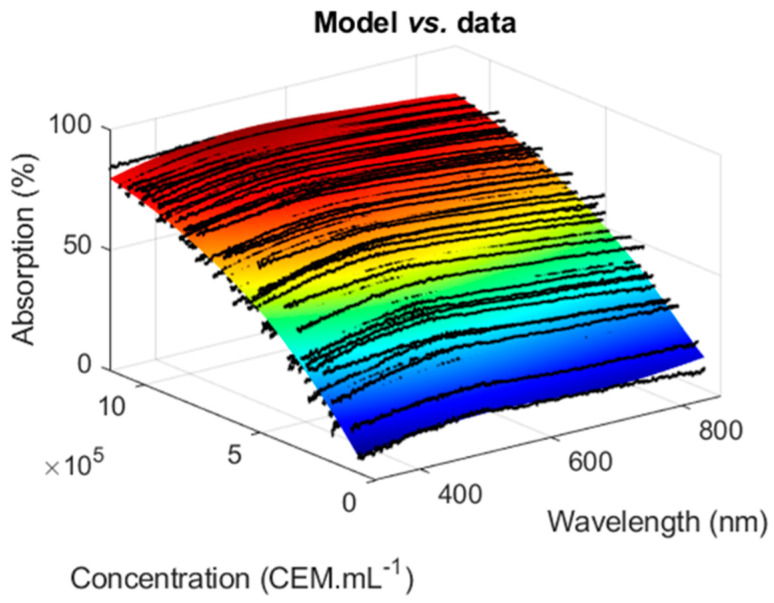
Description of the shape of CEM absorption spectra with a minimization algorithm. Black dot: experimental spectra, colored surface: Equation (5) plotted with parameters in Table 2.

**Figure 6 sensors-22-09223-f006:**
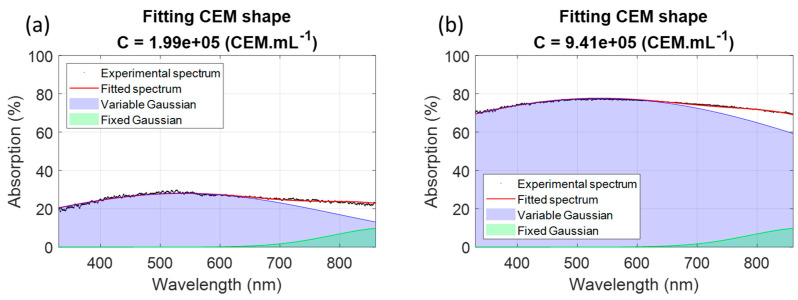
Examples of spectra fittings using Equation (5) and parameters in Table 2. (**a**) example at 1.99 × 10^5^ CEM.mL^−1^; (**b**) example at 9.41 × 10^5^ CEM.mL^−1^.

**Figure 7 sensors-22-09223-f007:**
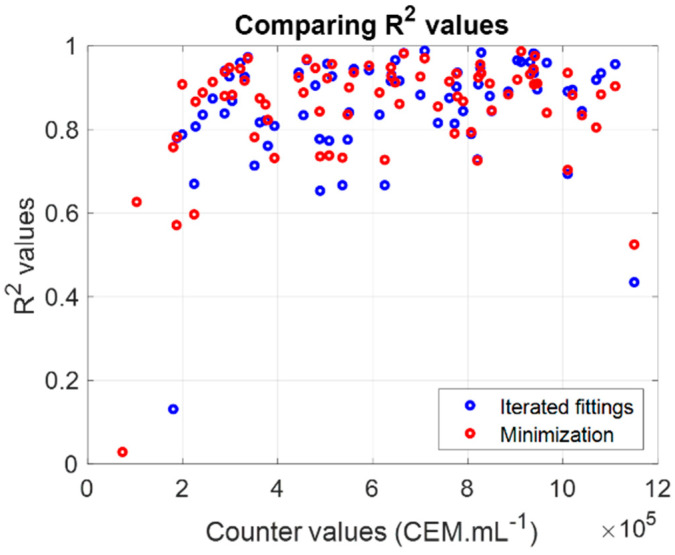
Comparing fitting efficiency with parameters issued from iterative fitting or minimization. Counter values: concentration measured with a cell counter from experimental cuvettes.

**Figure 8 sensors-22-09223-f008:**
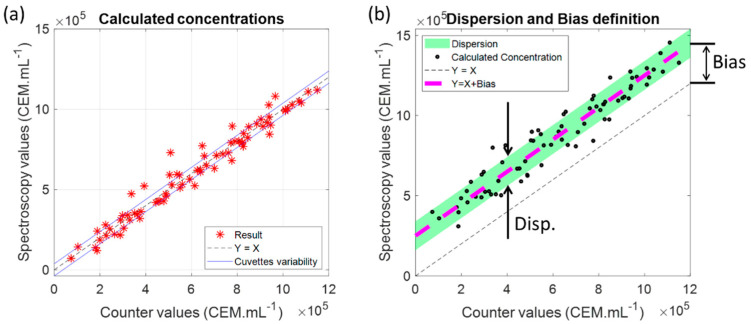
Measuring CEM concentrations using the CEM shape function. (**a**) experimental results; (**b**) definition of the Dispersion (Disp.) and Bias descriptors.

**Figure 9 sensors-22-09223-f009:**
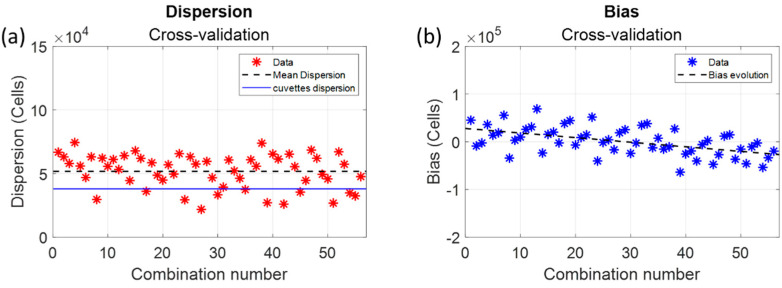
Dispersions and bias obtained using a cross-validation evaluation. (**a**) dispersion values; (**b**) bias values.

**Figure 10 sensors-22-09223-f010:**
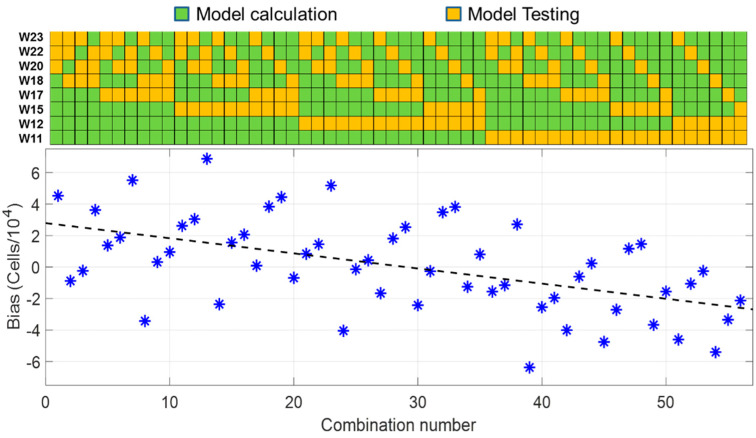
Evolution of the bias compared to sets used as « model » or « test » data. W(*n*) data in the figure refer to the weeks when CEM were grown and experimental data sets recorded. The eight data sets were recorded between March and June. Orange and green squares correspond to “model data” and “test data”, respectively.

**Figure 11 sensors-22-09223-f011:**
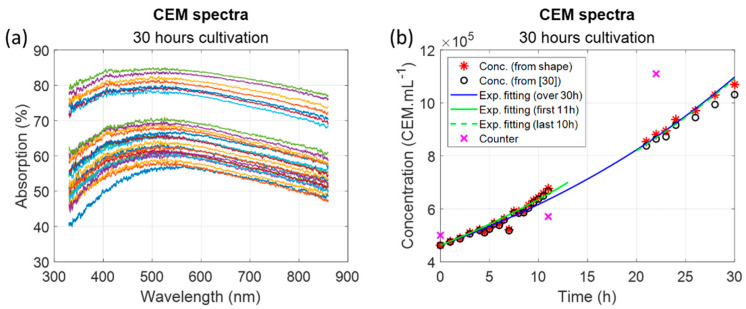
Monitoring CEM culture over 30 h. (**a**) recorded spectra; (**b**) calculated concentrations.

**Figure 12 sensors-22-09223-f012:**
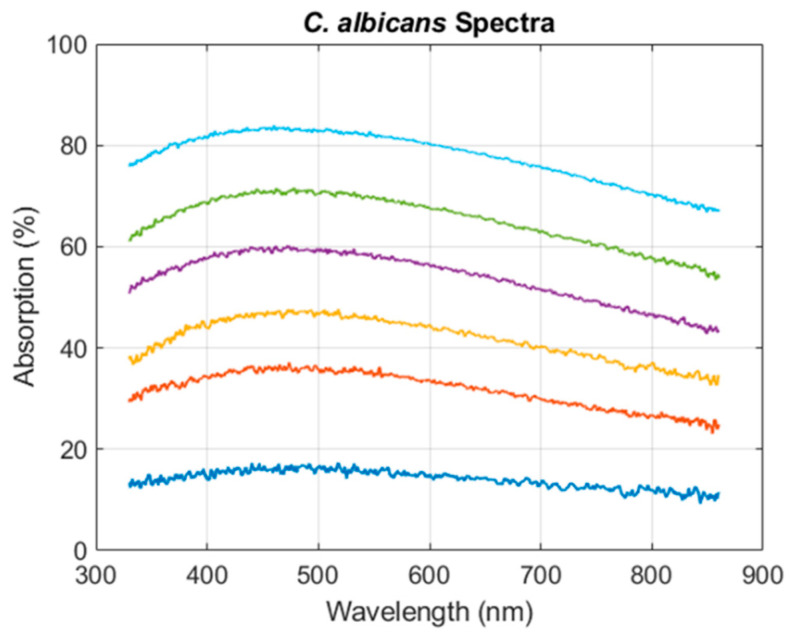
CA absorption spectra.

**Figure 13 sensors-22-09223-f013:**
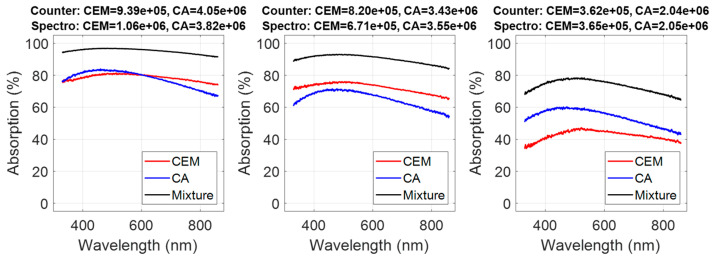
Measurements of co-culture concentrations of two species, CEM cells, and *Candida albicans.* The titles correspond to concentrations of both species measured by colony counting (Counter) and spectroscopy (Spec.).

**Figure 14 sensors-22-09223-f014:**
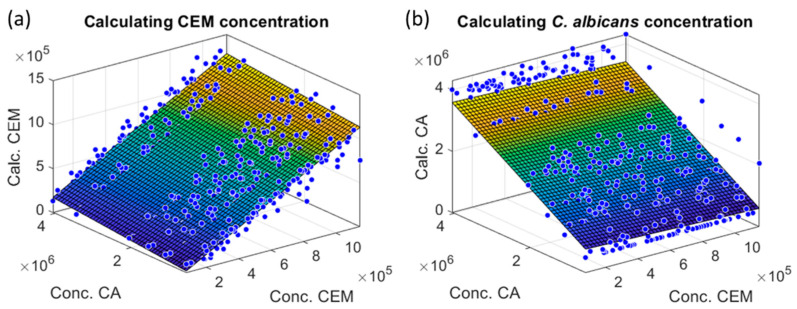
CEM and CA calculated concentrations as a function of initial species concentrations in the reconstructed spectra. Blue circles: data, colored surface: mean planes. (**a**) CEM; (**b**) CA.

**Table 1 sensors-22-09223-t001:** List of approximated parameters obtained by iterative fittings (CEM).

Approximated Parameters	p1a1	b1	p1c1	p2c1	a2	b2	c2
**Value**	7.45 × 10^−7^	496.9	2.14	0.41	23.11	976.9	253.9

**Table 2 sensors-22-09223-t002:** List of parameters obtained using a minimization algorithm (CEM).

Parameters	p1a1	b1	p1c1	p2c1	a2	b2	c2
**Value**	7.67 × 10^−7^	533.7	6.32	0.34	12.21	936.1	177.2

**Table 3 sensors-22-09223-t003:** List of parameters for *Candida albicans* function.

Parameters	p1a1	p2a1	p1b1	p2b1	c1	p1a2	p2a2	b2	c2
**Value**	−218.6	41.1	−7884	1130	2036	−182	33.36	562.3	479.5

**Table 4 sensors-22-09223-t004:** List of parameters of average planes.

Parameters	$p0_{CEM}$	$p1_{CEM}$	$p2_{CEM}$	$p0_{CA}$	$p1_{CA}$	$p2_{CA}$
**Value**	−7.5 × 10^4^	1.029	0.04	3.6 × 10^5^	−0.17	0.8

## Data Availability

Research data are available on demand from the corresponding author.

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
