# Peer review of "Absorption Spectra Description for T-Cell Concentrations Determination and Simultaneous Measurements of Species during Co-Cultures"

_sensors, 2022, doi:10.3390/s22239223_

Round 1

Reviewer 1 Report

The manusctript by Bruno Wacogne et.al. presented an optical method with white light spectroscopy aiming at measuring cell concentration in real time without sampling. The topic is interesting. There are several concerns should be addressed before accepted by Sensors publication. 

The author presented that the absorption spectra were collected for 8 weeks. How could you make sure that other parameters, except from cell concentration, would not affect the white spectra you measured during such a long time period of 8 weeks? For example the sedimentation of cell suspensions, cell proliferation etc.

why do the author use two gaussian curves to fit such "smooth"  absorption spectra? What is the logic or theory behind?

It seems that the "height" of spectra matters most in concentration determination rather than cell spectrum itself. what is the point of making sophisticated math model?

Is "an accuracy of 3.3%" sufficient for the cell density determination in a real application?

It seems not realized what the author claimed in the abstract that the method was able to determine concentrations of two species of cells. 

Author Response

Response to reviewer #1, manuscript N° 2029212

Authors would like to thank the reviewer for the attention paid to our manuscript.

  • Correction made are highlighted in red in the revised manuscript.
  • Please, get in touch with the editor to obtain the revised manuscript.

 Preliminary remark.

A typing error in our dispersion calculation Matlab™ programs has been detected. This has slightly changed some dispersion values. They have been corrected in the text (in red). Accordingly, figures 8(a) and 9(a) have been updated.

Reviewer’s comments and tentative answers.

  1. The author presented that the absorption spectra were collected for 8 weeks. How could you make sure that other parameters, except from cell concentration would not affect the white spectra you measured during such a long time period of 8 weeks? For example the sedimentation of cell suspensions, cell proliferation etc.

Cells were not used during an 8 weeks period. In order to generate a robust spectroscopy model, we need to have a large number of different concentrations (i.e. a large number of associated spectra). We managed to have almost 80 concentrations evenly (the best possible) distributed between 105 to 106 cell.mL-1. Since each spectroscopy measurement requires 2.5 mL of cell suspension, 8 cell cultures (one week per culture) were necessary. Each week, diluted cuvettes of 8 different concentrations distributed between 105 to 106 cell.mL-1 were prepared generating a continuum of concentrations.

Action made: section 2.1 has been modified to better explain the material and method regarding this issue.

  1. Why do the author use two gaussian curves to fit such "smooth" absorption spectra? What is the logic or theory behind?

This true that only 1 gaussian curve could have been enough to approximately describe the shape of the absorption spectra. However, spectra revealed the presence of a baseline. This baseline is described by the second and constant gaussian curve. Concentrations are measured by fitting the absorption spectra. Fitting the spectra with one or two gaussian curves is strictly equivalent in terms of calculation time. Using two gaussian curves lead to a higher R2, a better fitiing efficiency and a higher concentration determination accuracy; this is important for CEM cultures but crucial when considering co-cultures of several species, especially when one of the species exhibits high absorption around 800 nm wavelength.

Action made: a sentence concerning this remark has been added in section 4.3.

  1. It seems that the "height" of spectra matters most in concentration determination rather than cell spectrum itself. What is the point of making sophisticated math model?

This is true and this is pointed out in section 4.3 where we explain that the evolution of  is described with an exponential function of the same form as the one already used in the Beer-Lambert derived method presented in [30]. However, concentration measurement method relying on the estimation of only one parameter (Beer-Lambert derived methods or cell counters) cannot be used to simultaneously monitor concentrations of several species. For this, multi-parameters method should be employed, among them spectroscopy.

Action made: This remark has been added in the introduction of the paper.

  1. Is "an accuracy of 3.3%" sufficient for the cell density determination in a real application?

Measuring cell concentration with 3% accuracy is as accurate (probably more accurate) than what is obtained nowadays with conventional techniques. Indeed, counting cells with a Malassez cell leads to accuracies between 10-20 % and is most of the time operator dependent. Commercial cell counters are very accurate but they require very small volumes (10-20 µL) which are hardly representative of the content of the suspension under investigation. We measured a dispersion of about 10% (and not 5% as written, value corrected) with an automated cell counter (reference [30] of the paper). As mentioned in the text, “an accuracy about 20% is still acceptable (personal communication with the French Blood Agency)”.

Action made: a paragraph has been added to section 4.4 to summarize these comments.

  1. It seems not realized what the author claimed in the abstract that the method was able to determine concentrations of two species of cells.

Results shown in section 3.2.2 were obtained using reconstructed mixture spectra using equation (10). We may not have been clear enough when using the word “reconstructed” throughout the text. Co-cultures with CEM and Escherichia coli are currently being studied and an accuracies of about 3% for both concentrations measurements are observed.

Action made: a paragraph has been added to section 4.6 to recall that the illustration of co-culture monitoring was made with reconstructed spectra.

Reviewer 2 Report

This paper makes a bold and innovative attempt in the cell concentration system, using optical and data fitting to observe cell concentration and try to test the concentration of two organisms in a co-culture system, although the accuracy is not high, but it is a novel attempt. It can be seen that the article did a lot of work and complex calculations to obtain the data and organize the fitted data. The idea is a very forward-thinking. However, there are some minor issues that have not been resolved.

1. ATMP is mentioned in the abstract and introduction of the article, but the space it occupies is not important relative to the entire article.

2. The experiment and data calculation are not precise enough, the references of the equation are not specific enough, and the algorithm is not clear.

3. The accuracy of the co-culture system is indeed too low, only 3%, it is recommended to improve the accuracy.

4. The degree of discreteness of individual data is too large, whether there is a possibility that the experimental data cannot be repeated.

5. Pictures are duplicated, some 3D maps do not provide more useful data, slightly duplicated.

Author Response

Response to reviewer #2, manuscript N° 2029212

Authors would like to thank the reviewer for the attention paid to our manuscript and the kind introduction paragraph.

  • Correction made are highlighted in red in the revised manuscript.
  • The revised manuscript is uploaded together with our answers.

Preliminary remark.

A typing error in our dispersion calculation Matlab™ programs has been detected. This has slightly changed some dispersion values. They have been corrected in the text (in red). Accordingly, figures 8(a) and 9(a) have been updated.

Concerning the English correction.

This is the major modification required.

We acknowledge that we are not native English speakers. Up to now, our English expression was tolerated but we fully accept this remark.

Action made: the paper has been completely corrected by a native English colleague.

Reviewer’s comments and tentative answers.

  1. ATMP is mentioned in the abstract and introduction of the article, but the space it occupies is not important relative to the entire article.

The work presented in this paper was performed in the frame of a much more global project in which we work on T-cell culture monitoring using white light spectroscopy. ATMP fabrication is therefore the general context but not the goal of the paper which is to present one of the spectroscopy methods we set-up during this project. It is true that ATMP is mentioned in the abstract (2 lines over 10 of the abstract) and in the introduction (17 lines over 693 lines of the paper) which represents a very small part of the manuscript but could make a reader think that the entire paper deals with the fabrication of ATMP.

We could further reduce mentions to ATMP but this could make the description of the context a bit poor. Should we do it or change some parts of the abstract or introduction to specify that our study concerns a method to monitor cell culture and that ATMP is one of the possible applications?

  1. The experiment and data calculation are not precise enough, the references of the equation are not specific enough, and the algorithm is not clear.

Concerning precision.

We think this is in line with comment 3, we address it below.

Concerning references of the equations.

In this paper, theoretical developments and consequent equations are the results of our own investigations and not equations issued from the literature. Indeed, the goal of the paper is precisely to establish these equations. This is true for all equations except equation (2) which is the direct application of the absorbance additivity law and equation (1) which is a simple definition.

Is this answer correct or have we misunderstood the reviewer’s comment?

Concerning the algorithm.

We understand this remark concerns the MATLABTM “minimization algorithm” we used to calculate the function coefficients. It would be long to explain how the minimization algorithm works and this would lengthen the paper (to our opinion). However, we understand that a reference to the corresponding MATLABTM documentation website is missing.

Action made: we add the MATLABTM webpage address where documentation on this algorithm can be found between brackets in section3.1.2.

  1. The accuracy of the co-culture system is indeed too low, only 3%, it is recommended to improve the accuracy.

3% is the dispersion observed during the 30 hours cultivation experiment and not while investigating reconstructed co-cultures. Measuring cell concentration with 3% accuracy is as accurate (probably more accurate) than what is obtained with conventional techniques. For example, counting cells with a Malassez cell leads to accuracies between 10-20 % and is human dependent. Commercial cell counters are very accurate but they require very small volumes (10-20 µL) which are hardly representative of the content of the suspension under investigation. We measured a dispersion of about 10% (and not 5% as written, value corrected) with an automated cell counter (reference [30] of the paper). As mentioned in the text, “an accuracy about 20% is still acceptable (personal communication with the French Blood Agency)”.

The 10 % dispersion observed in figure 9(a) is due to the variability of the optical properties of basic plastic spectroscopy cuvettes. This is mentioned in section 4.4 while referring to an ancillary study not shown in this paper. Therefore, the dispersion observed in figure 9(a) is due to the cuvette variability and not to the spectroscopy method we propose which is about 3% as measured in section 3.1.4. As a consequence, co-culture reconstruction suffers from this twice: dispersion of the CEM cuvettes and dispersion of the yeast cuvettes.

Therefore, the accuracy of the method is good (about 3%) as it is meant to be used for monitoring purpose.

Action made: a paragraph has been added to section 4.4 to summarize these comments.

  1. The degree of discreteness of individual data is too large, whether there is a possibility that the experimental data cannot be repeated.

We understand the origin of this misunderstanding which is due to our way to present section 2.1.

Usually, experiments are performed in duplicate or triplicate to check the accuracy of experiments. Reading the paper, it can be unfortunately understood that we did not conducted 2 or 3 sets of experiments but 8. It was not the case. In order to generate a robust spectroscopy model, we need to have a large number of different concentrations (i.e. a large number of associated spectra). Using 8 times identical data would have resulted in a less robust model. We managed to have almost 80 concentrations evenly (the best possible) distributed between 105 to 106 cell.mL-1. Since each spectroscopy measurement requires 2.5 mL of cell solution, 8 weeks of cell culture were necessary. Each week, diluted cuvettes of 8 different concentrations distributed between 105 to 106 cell.mL-1 were prepared generating a continuum of concentrations.

Action made: section 2.1 has been modified to avoid this misunderstanding.

  1. Pictures are duplicated, some 3D maps do not provide more useful data, slightly duplicated.

We agree.

Action made: figures 2(b) and 12(b) have been removed and captions modified.

Reviewer 3 Report

Wacogne et al. presented "Absorption spectra description for T-cell concentrations determination and simultaneous measurements of species during co-cultures". White light spectroscopy was employed in this study to calculate T-cell concentrations based on the form of the absorption spectra of various cell solution dilutions. This technique determined cell concentration with an accuracy of roughly 3% during a 30 hour cultivation monitoring and a doubling period of 26 hours. Furthermore, the described approach allowed concentrations of two species to be determined inside recreated co-cultures of T-cells and Candida albicans yeasts.  The presented method can enable real-time measurements. The core-idea of the manuscript can be potentiall useful for quality control during cell production.  The manuscript is fluent and the data support the claims of the authors. I have the following comments.

*Figures should be revised to increase the text size of the labels and insets. 

*is there any ptoential approach that can be used to improve the accuracy of the presented method?

*What is the effect of sedimentation? Did the authors abserve any effect of sedimentation?

Author Response

Response to reviewer #3, manuscript N° 2029212

 Authors would like to thank the reviewer for the attention paid to our manuscript.

  • Correction made are highlighted in red in the revised manuscript.
  • The revised manuscript is uploaded together with our answers.

Preliminary remark.

A typing error in our dispersion calculation Matlab™ programs has been detected. This has slightly changed some dispersion values. They have been corrected in the text (in red). Accordingly, figures 8(a) and 9(a) have been updated.

Reviewer’s comments and tentative answers.

  1. Figures should be revised to increase the text size of the labels and insets.

We suppose this remark concerns multiple figures and not single figures.

Action made: Text size has been increased in figures 3, 6, 8, 9, 11, 13 and 14.

  1. Is there any potential approach that can be used to improve the accuracy of the presented method?

Indeed, measuring concentration with our method is accurate (3% measured in section 3.1.4.). For example, counting cells with a Malassez cell leads to accuracies between 10-20 % and is human dependent. Commercial cell counters are very accurate but they require very small volumes (10-20 µL) which are hardly representative of the content of the suspension under investigation. We measured a dispersion of about 10% (and not 5% as written, value corrected) with an automated cell counter (reference [30] of the paper). As mentioned in the text, “an accuracy about 20% is still acceptable (personal communication with the French Blood Agency)”.

The 10 % dispersion observed in figure 9(a) is due to the variability of the optical properties of basic plastic spectroscopy cuvettes. This is mentioned in section 4.4 while referring to an ancillary study not shown in this paper. Therefore, the dispersion observed in figure 9(a) is due to the cuvette variability and not to the spectroscopy method we propose which is about 3% as measured in section 3.1.4. As a consequence, co-culture reconstruction suffers from this twice: dispersion of the CEM cuvettes and dispersion of the yeast cuvettes.

Action made: a paragraph has been added to section 4.4 to summarize these comments.

  1. What is the effect of sedimentation? Did the authors observe any effect of sedimentation?

We did not observe any effect of sedimentation because suspensions were homogenized by several gentle inversions before each spectroscopy measurement.

Action made: we specified this in section 2.4.

Round 2

Reviewer 2 Report

Accepted